# Multi-Objective Evolutionary Architecture Search for Parameterized Quantum Circuits

**DOI:** 10.3390/e25010093

**Published:** 2023-01-03

**Authors:** Li Ding, Lee Spector

**Affiliations:** 1Manning College of Information & Computer Sciences, University of Massachusetts Amherst, Amherst, MA 01002, USA; 2Department of Computer Science, Amherst College, Amherst, MA 01002, USA

**Keywords:** quantum computing, quantum machine learning, reinforcement learning, evolutionary algorithms, multi-objective optimization

## Abstract

Recent work on hybrid quantum-classical machine learning systems has demonstrated success in utilizing parameterized quantum circuits (PQCs) to solve the challenging reinforcement learning (RL) tasks, with provable learning advantages over classical systems, e.g., deep neural networks. While existing work demonstrates and exploits the strength of PQC-based models, the design choices of PQC architectures and the interactions between different quantum circuits on learning tasks are generally underexplored. In this work, we introduce a Multi-objective Evolutionary Architecture Search framework for parameterized quantum circuits (MEAS-PQC), which uses a multi-objective genetic algorithm with quantum-specific configurations to perform efficient searching of optimal PQC architectures. Experimental results show that our method can find architectures that have superior learning performance on three benchmark RL tasks, and are also optimized for additional objectives including reductions in quantum noise and model size. Further analysis of patterns and probability distributions of quantum operations helps identify performance-critical design choices of hybrid quantum-classical learning systems.

## 1. Introduction

Noisy Intermediate-Scale Quantum (NISQ) technology [1] reveals the potential of quantum systems to have significant and reliably advantage in solving computing tasks over classical systems. One of the most encouraging areas of quantum computing research is hybrid quantum-classical learning systems, which utilizes parameterized quantum operations together with classical optimization methods to solve the designed learning tasks. In particular, a standard way to model hybrid systems is to use parameterized quantum circuits (PQCs) [2]. Recent work has demonstrated that PQC-based systems are able to handle a variety of supervised and unsupervised tasks such as classification [3,4,5,6] and generative modeling [7,8,9,10]. Moreover, some recent work [11,12] further demonstrates that PQCs can be used to construct parameterized quantum policies to solve more complex reinforcement learning problems, and shows provable learning advantages in some specific RL environments over classical methods such as deep neural networks (DNNs).

An essential element of PQC-based systems is the design and structural arrangement of quantum and classical operations. Prior work [11,12,13] has shown significant impact of utilizing some key components, as well as analyzed different design choices through empirical studies. However, the development of best-performing architectures of hybrid quantum learning systems still relies on human ingenuity. In classical learning systems, architecture search methods that aim to automate the process of discovering and evaluating the architecture of complex systems have been extensively explored in the past decade, e.g., neural architecture search (NAS) [14]. In the context of quantum computing, while recent work [15] made early attempts to apply evolutionary NAS methods [16] in searching for PQC architectures, their method does not fully consider the interactions between quantum gates, which results in much complexity and computational redundancy. Overall, automated search and optimization of architectures of hybrid quantum learning systems have not been sufficiently explored.

In this work, we aim to explore using multi-objective genetic algorithms [17] to automatically design the architecture of hybrid quantum-classical learning systems that can solve complex RL problems. We start by adopting the ideas of successful approaches in NAS and QAS using genetic algorithms [15,16], which have more flexible architecture search spaces and require less prior knowledge than other gradient-based approaches such as RL. We propose MEAS-PQC, a Multi-objective Evolutionary Architecture Search framework for PQC-based quantum learning models, which uses a multi-objective genetic algorithm with quantum-specific configurations to perform efficient searching of optimal PQC architectures. More specifically, our method improves prior work [15] in three ways: (1) we adopt the Uniform Mutation by Addition and Deletion (UMAD [18]) mutation operator to enable searching for architectures with variable lengths; (2) our method specifically considers equivalent quantum operations in the architecture and eliminates unnecessary redundancy; (3) we further extend quantum architecture search to a more practical setting by forming a multi-objective optimization problem, which considers specific factors in real-world quantum computing, e.g., interactions between quantum gates and quantum noise, in addition to learning performance.

To validate the proposed method, we implement our method on three benchmark RL environments from OpenAI Gym [19], which has been extensively used in RL research. The experiments consider both single- and multi-objective settings. For single-objective, our method outperforms prior work by a significant margin on learning performance, and we also perform qualitative analysis on the results to further interpret and discover the essential design choices of PQC-based systems. For multi-objective, we observe that our method is able to search for architectures that have good learning performance as well as meet computational constraints such as quantum noise and model size.

To summarize, the contributions of this work are two folds: (1) we propose MEAS-PQC, which improves prior work [15] by adding performance-critical components including UMAD mutation and simplification of equivalent quantum gates; (2) we form quantum architecture search as a multi-objective optimization problem, and shows that our method can search for PQC architectures that not only have good learning performance, but are also more efficient and robust to potential quantum noise.

## 2. Preliminaries and Related Work

In this section, we introduce some basic concepts of quantum computation that are closely related to this work, and give a detailed description of parameterized quantum circuits and other related work.

### 2.1. Quantum Computation Preliminaries

A complex Hilbert space of $2^{n}$ dimensions usually serves a general representation of a multi-qubit quantum system with *n* qubits. The quantum state of the system is mathematically written as a vector |ψ〉, which has unit norm 〈ψ|ψ〉=1. 〈ψ| is the conjugate transpose and 〈ψ|ψ〉 represents the inner-product under the bra-ket notation. We represent the computational basis states of a multi-qubit system as tensor-products of single-qubit basis states. For example, a two-qubit state |01〉=|0〉⊗|1〉, where $|0\rangle =\left[\begin{array}{c} 1\\ 0 \end{array}\right]$ and $|1\rangle =\left[\begin{array}{c} 0\\ 1 \end{array}\right]$ are two single-qubit basis states.

A quantum gate is mathematically represented as a unitary operator *U* acting on qubits. In quantum machine learning, a few common quantum gates are extensively used, e.g., the single-qubit rotation operators $R_{x}$, $R_{y}$, $R_{z}$. Given a rotation angle θ, the matrix representations of rotation operators are
(1)$$R_{x}\left(\theta \right)=\left[\begin{array}{cc} \cos\frac{\theta }{2} & -i\sin\frac{\theta }{2}\\ -i\sin\frac{\theta }{2} & \cos\frac{\theta }{2} \end{array}\right],R_{y}\left(\theta \right)=\left[\begin{array}{cc} \cos\frac{\theta }{2} & -\sin\frac{\theta }{2}\\ \sin\frac{\theta }{2} & \cos\frac{\theta }{2} \end{array}\right],R_{z}\left(\theta \right)=\left[\begin{array}{cc} e^{-i\frac{\theta }{2}} & 0\\ 0 & e^{i\frac{\theta }{2}} \end{array}\right].$$

An *entangled* state (or *entanglement*) is a quantum state of a composite system that can not be written as a tensor-product of the states of its components. An entanglement can be created by applying appropriate 2-qubit gates. In this work, we mainly use controlled-Pauli-Z gates to generate entanglement. The Controlled-Z gate (CZ) is a symmetric gate,
(2)CZ=|0〉〈0|⊗I+|1〉〈1|⊗Z

A projective measurement of quantum states is described by an observable, *M*, which is a Hermitian operator on the state space of the quantum system being observed. The observable has a spectral decomposition
(3)$$M=\sum_{m}mP_{m},$$
where $P_{m}$ is the projector onto the eigenspace of *M* with eigenvalue *m*. Upon measuring the state |ψ〉, the probability of getting result *m* is given by
(4)$$p\left(m\right)=\langle \psi |P_{m}|\psi \rangle ,$$
and the expectation value of the measurement is
(5)$$E\left(M\right)=\sum_{m}m\cdot p\left(m\right)=\langle \psi |M|\psi \rangle .$$

For a more detailed introduction to quantum computation and its basic concepts, we refer the readers to Nielsen and Chuang [20].

### 2.2. Parameterized Quantum Circuits

Given a fixed *n*-qubit system, a parameterized quantum circuit (PQC) is defined by a unitary operation U(s,θ) that acts on the current quantum states *s* considering the trainable parameters θ. In this work, we mainly consider two types of PQCs: variational PQCs (V-PQCs) [2,21] and data-encoding PQCs (D-PQCs) [13,22], which are widely used in quantum machine learning. The V-PQCs are composed of single-qubit rotations $R_{x}$, $R_{y}$, $R_{z}$ with the rotation angles as trainable parameters. Similar to classical learning models, the PQC parameters can also be optimized with gradient-based algorithms. The task of learning is mathematically expressed as the minimization of a defined objective function L(θ) with respect to the parameter θ of a V-PQC. The gradient descent algorithm updates θ towards the direction of descending the loss
(6)$$\theta \leftarrow \theta -\eta \nabla_{\theta }L,$$
where η is the learning rate and $\nabla_{\theta }L$ is the gradient vector. The gradient can be estimated through a finite difference of partial derivatives
(7)$$\frac{\partial L}{\partial \theta_{i}}\approx \frac{L\left(\theta +\Delta e_{i}\right)-L\left(\theta -\Delta e_{i}\right)}{2\Delta }$$
where Δ is a small constant and $e_{i}$ is the Cartesian unit vector in the *i* direction.

The D-PQCs have a similar structure with rotations, but the angles are the input data *d* scaled by a trainable parameter λ. The input data are usually normalized to have similar scales. For example, in our experiments, the input data are state variables in the RL environments, and each of them is normalized according to its min-max values. The structures of both PQCs are depicted in Figure 1, which we describe in detail later in Section 3.2.

Recent work [11] proposes to use an alternating-layered architecture [13,22] to implement parameterized quantum policies for RL, which basically applies an alternation of V-PQC (followed by an entanglement) and D-PQC till the target depth, while this architecture is simple and effective, it is obvious to see that this general design can be easily modified and probably improved by changing the placement of its components. In this work, we aim to optimize the design of such PQC-based systems with architecture search methods.

### 2.3. Quantum Architecture Search

Early research [23] has shown the usage of genetic programming to solve specific quantum computing problems from an evolutionary perspective. Common architecture search approaches, such as greedy algorithms [24,25], evolutionary algorithms [26,27,28,29], reinforcement learning [30,31], and gradient-based learning [32] have also been attempted to solve tasks such as quantum control, variational quantum eigensolver, quantum error mitigation, and entanglement purification. However, most of these approaches target optimizing either specific pure quantum circuits or single-qubit quantum operations, instead of more complex multi-qubit hybrid systems.

More recently, a few approaches have been proposed to optimize the architectures involving parameterized quantum circuits. Grimsley et al. [24] proposed a method that iteratively adds parameterized gates and re-optimizes the circuit using gradient descent. Ostaszewski et al. [33] proposed an energy-based searching method for optimizing both the structure and parameters for single-qubit gates and demonstrated its performance on a variational quantum eigensolver. Ding and Spector [15] proposed EQAS-PQC, which applies existing evolutionary NAS method to perform search on PQC architectures, while EQAS-PQC shows promising and encouraging results, their method suffers considerable computational redundancy in the searched architectures. In this work, we take steps further and propose a more general multi-objective architecture search framework that is specialized for hybrid quantum-classical systems.

## 3. Method

We propose MEAS-PQC, an evolutionary framework that uses multi-objective genetic algorithms to perform quantum architecture search for PQC-based hybrid quantum-classical systems. In this section, we describe the major components of MEAS-PQC including the genetic algorithm, encoding scheme, search process, and the multi-objective optimization framework in detail.

### 3.1. Genetic Algorithm for Quantum Architecture Search

Genetic algorithms (GAs) refer to a class of population-based computational paradigms, which simulate the natural evolution process to evolve programs by using artificial genetic operations (e.g., crossover and mutation) to optimize fitness or objective functions. It has been successfully adopted in recent work for optimizing complex systems such as neural networks [16,34,35]. In the perspective of GAs, we view the architectures of quantum circuits as *phenotypes*, and define representations as *genotypes* on which the genetic operations can be easily applied. Similar to many other genetic algorithms, MEAS-PQC iteratively generates a population of candidates (architectures) through genetic operations on the given parents, and selects parents for the next generation based on fitness evaluation.

In this work, we adopt the Non-Dominated Sorting Genetic Algorithm II (NSGA-II) [36] to optimize the search process. NSGA-II is an evolutionary algorithm that has been successfully employed in various single- and multi-objective optimization problems including NAS [16]. The algorithm has the following procedures:1.Perform a non-dominated sorting in the population of quantum architectures and classify them according to an ascending level of non-domination based on objectives.2.Use crowding distance, which is related to the density of solutions with similar objective metrics, to perform Crowding-sort that makes the population less dense.3.Generate offspring using crowded tournament selection, then apply genetic operators such as mutation and crossover.

In the context of this work, for single-objective optimization, MEAS-PQC generates a population of PQC architectures in the search space, and iteratively selects the ones with good learning performance. For multi-objective, it uses an elitist principle to select the non-dominated candidates using the above procedure. The benefit of using NSGA-II is that it automatically selects a good set of quantum architectures while balancing different objectives. For multi-objective optimization, one naive way is to sum all the objectives numerically, which requires to carefully weigh different objectives to avoid dominance of objectives that have large value scale. Instead, our method does not need to perform such tuning, which is more efficient and robust.

### 3.2. Encoding Scheme and Search Space

An encoding scheme is the interface for abstracting the architectures to genomes, where the genes are representations for different quantum operations. We follow the general encoding scheme as proposed in Ding and Spector [15], which consists of some basic functional quantum circuits on a single qubit or multiple qubits that has been widely adopted in prior work [11]. An illustration of a simple PQC architecture in the search space is depicted in Figure 1. More specifically, we define four basic operation encodings $x=\{x_{0},x_{1},x_{2},x_{3}\}$, and the corresponding genes are represented as integers {0,1,2,3}. Given a fixed *n*-qubit state, we define the following operations:$x_{1}$: Variational PQC—A circuit with single-qubit rotations $R_{x}$, $R_{y}$, $R_{z}$ (Equation (Equation 1)) performed on each qubit, with the rotation angles as trainable parameters. For generality, we consider qubit rotation in 3-dimensional space (i.e., applying $R_{x}$, $R_{y}$, $R_{z}$) for all the qubits. This operator is used to change the qubit states based on the trainable parameters.$x_{2}$: Entanglement—A circuit that performs circular entanglement to all the qubits by applying one or multiple controlled-Z gates (Equation (Equation 2)). In this work, we only consider circular entanglement, which has been widely used in prior studies.$x_{3}$: Data-encoding PQC—A circuit with single-qubit rotations $R_{x}\left(\theta \right)$ (Equation (Equation 1)) performed on each qubit, with the rotation angle θ being the input data *d* scaled by trainable parameters λ,
(8)θ=σ(λd)
where σ(·) is the activation function. Similar to $x_{1}$, this operator is also used to change the qubit states, but based on scaled/transformed input values, which is how the input data takes effect.While prior work Ding and Spector [15] uses linear activation, we propose to use non-linear activations such as Tanh to allow effective stacks of consecutive $x_{3}$ circuits, since linear activation leads to multiple consecutive rotations being equivalent to a single rotation (Equation (Equation 11)).$x_{0}$: Measurement—A Variational PQC ($x_{1}$) with trainable parameters followed by measurement to obtain the qubit observables. The outcome is a binary value for each qubit with different probabilities. The outputs are computed by a linear weighting of the observables by another set of trainable parameters for each output, with optional activation functions, e.g., Softmax for action probabilities. The architecture encoding/decoding is terminated when approaching $x_{0}$.

The search space of MEAS-PQC depends on the maximal length of the genomes. Since the encoding will terminate when approaching to $x_{0}$, all the genomes will have only one $x_{0}$ at the end. So the search space is the sum of possible operations (except $x_{0}$) for all the possible lengths less than the maximum length. Given a maximum length of the genomes *n*, the search space of our method is
(9)$$\Omega_{x,n}=\sum_{i=1}^{n}{\left(|x|-1\right)}^{i-1}-K$$
where *K* is the number of duplicate genomes that can be decoded to equivalent architectures, which is described in detail in the following section.

### 3.3. Evolutionary Quantum Architecture Search

MEAS-PQC is proposed to generate diverse sequential combinations of quantum operators and iteratively search for good candidates with respect to the objectives. The search process is an evolutionary algorithm, and we elaborate on the following key components:

*Mutation.* To efficiently scan the search space for better genomes, we use Uniform Mutation by Addition and Deletion (UMAD [18]). UMAD is a mutation method that performs addition and deletion of genes separately. It has shown superior performance in many evolutionary computing applications such as program synthesis. For each mutation operation, UMAD deletes/adds genes to the genome controlled by addition/deletion rate *r*. Given genome of length *l*, there are *l* positions for deletion and l+1 positions for addition (including start and end). We perform size-neutral UMAD with
(10)$$r_{del}=1/l,r_{add}=1/\left(l+1\right)$$
so that the expected number of added/deleted genes is 1 after each mutation.

Comparing to traditional mutation operators, such as polynomial mutation that has been used in prior QAS work [15], UMAD has two major advantages. First, the traditional mutation operator replaces a randomly chosen gene with a new, randomly generated gene, and thus the length of genome does not change. In our case, we expect the algorithm to search for architectures with different lengths based on the parent, which can be achieved by UMAD. Secondly, since UMAD dissociates the additions from the deletions, it can thus provide more paths through the search space from a parent to a descendant, especially when specific genes are essential to the performance [18].

*Equivalence of Quantum Operators.* It is worth noting that some architectures in the search space are computationally equivalent, as some quantum operations can be linearly combined. More specifically, given two rotation operators on the same axis $R_{x}\left(\theta_{1}\right)$, $R_{x}\left(\theta_{2}\right)$ (x-axis for instance), from Equation (Equation 1), we can derive that
(11)$$R_{x}\left(\theta_{1}\right)\cdot R_{x}\left(\theta_{2}\right)=\left[\begin{array}{cc} \cos\frac{\theta_{1}}{2}\cos\frac{\theta_{2}}{2}-\sin\frac{\theta_{1}}{2}\sin\frac{\theta_{1}}{2} & -i\cos\frac{\theta_{1}}{2}\sin\frac{\theta_{2}}{2}-i\cos\frac{\theta_{2}}{2}\sin\frac{\theta_{1}}{2}\\ -i\cos\frac{\theta_{1}}{2}\sin\frac{\theta_{2}}{2}-i\cos\frac{\theta_{2}}{2}\sin\frac{\theta_{1}}{2} & \cos\frac{\theta_{1}}{2}\cos\frac{\theta_{2}}{2}-\sin\frac{\theta_{1}}{2}\sin\frac{\theta_{1}}{2} \end{array}\right].$$

Let $\theta_{3}=\theta_{1}+\theta_{2}$, we have $R_{x}\left(\theta_{3}\right)=R_{x}\left(\theta_{1}\right)\cdot R_{x}\left(\theta_{2}\right)$. In other words, two consecutive rotations can be equivalently replaced by one rotation. As a result, if the genome has two or more consecutive $x_{1}$, we can reduce the genome by keeping only one $x_{1}$.

Similarly, given Equation (Equation 2), we have
(12)$$CZ\cdot CZ=\left[\begin{array}{cc} I & 0\\ 0 & I \end{array}\right],$$
which means two consecutive $x_{2}$ becomes actually identity, which can be safely removed from the genome. We perform duplicate elimination at each generation based on these equivalence of quantum operators to avoid the redundancy in the population.

### 3.4. Multi-objective Optimization for Quantum-Classical Systems

In this work, we consider three objectives for PQC-based learning systems: learning performance, quantum noise, and model size. First, the main objective is the learning performance. For each generation, we decode the population to different architectures, and use the architectures to construct PQC policies for the RL agents. The learning performance is computed as the average episode reward in the target RL environment to represent the area under the learning curve.

We also consider two other objectives that are essential to real-world applications of quantum systems: quantum noise and model size. Quantum noise usually refers to unexpected (and typically unwanted) random variation due to the discrete nature of photons [37]. The effect of quantum noise that occurs throughout a computation process can be quite complex, as there are many potential causes, e.g., thermal fluctuations, mechanical vibrations. While some recent work [38] reported the system-wide Pauli and measurement errors on specific quantum hardware regarding different numbers of qubits, the errors of different quantum gates and operations have not been found in past empirical studies. As a result, given no knowledge of the actual quantum hardware, we propose a rough measure of the theoretical quantum noise in PQC based systems.

Given a system with *n* qubits in the MEAS-PQC search space with operations $x_{i}$, i∈0,1,2,3, we define the system-level noise
(13)$$\Delta =n\cdot (3\sum_{i=0,1}\alpha_{1}\cdot \#x_{i}+2\alpha_{2}\cdot \#x_{2}+\alpha_{3}\cdot \#x_{3}).$$
In other words, we assume that the total quantum noise is proportion to the total number of quantum gates operating on any qubit. For this work, we take an artificial setting of $\alpha_{i}=1,i\in 1,2,3$, meaning different gates are considered to have the same level of quantum noise. Notably, this setting can be further refined for specific quantum hardware.

For any parameterized learning models, model size, or number of parameters, is always an important factor that is closely related to computation cost, memory, storage, and other computation-related requirements. For our method, we also calculate the total number of parameters as
(14)$$N=n\cdot (3\cdot \sum_{i=0,1}\#x_{i}+\#x_{3}).$$
Note that since the search space increases exponentially to the maximum length of the architecture (Equation (Equation 9)), it is important to add penalty to the model size when searching for PQC architectures.

## 4. Experiments

In this section, we describe the implementation details and experimental results of MEAS-PQC on the benchmark RL environments regarding both single- and multi-objective optimization. We also compare our method to prior work to demonstrate the advantage of it against commonly-used alternating-layer PQC architectures (Softmax-PQC by Jerbi et al. [11]) and classic genetic algorithms (EQAS-PQC by Ding and Spector [15]).

### 4.1. RL Environments

In this work, we consider three classical RL benchmark environments from the OpenAI Gym [19]: CartPole, MountainCar, and Acrobot, which have been widely used in RL research, including prior work on quantum RL [11,12,15]. The CartPole task is to prevent the pole from falling over by controlling the cart. For MountainCar, the goal is to drive up the mountain by driving back and forth to build up momentum. Acrobot refers to a swing-up task, in which the system must use the elbow (or waist) torque to move the system into a vertical configuration then balance. Detailed description can be found in Brockman et al. [19].

The specifications for RL environments are presented in Table 1, where the reward is the step reward and γ is the discount factor for future rewards. We follow prior work to use a single qubit to represent each state variable, while it is hard to interpret the meaning of qubit states, the D-PQC rotates the qubit based on the value of the state variable. In the scope of this work, we do not further optimize the number of qubits as a hyperparameter, but it can certainly be done with the proposed method, by including the size of qubits as a variable in the search space.

### 4.2. Implementation Details

For all the experiments, we perform quantum computation via noiseless simulation. The quantum circuits are implemented using Cirq [40] and TensorFlow Quantum [41]. The main search process is implemented using the *pymoo* [42] framework. Other specifications and hyper-parameters for architecture search and RL training are described as follows.

*Architecture Search.* MEAS-PQC uses NSGA-II with a population size of 30 and runs for 50 generations. The maximum length of architectures is set to 20, i.e., the largest architecture will have 20 consecutive operations varying from $x_{1}$ to $x_{3}$, followed by measurement $x_{0}$. For each generation, we evaluate the learning performance of the population by running 5 trials in the RL environments with a reduced number of episodes to a factor of 0.5, which improves the efficiency of evolution. We also compute the estimation of quantum noise and computational cost as described in Section 3.4. For the final results, we evaluate the architectures produced by our method as well as prior work for 20 trials in order to reduce the variance caused by inevitable failure cases due to bad initialization of parameters and environment states.

*RL training during search.* We set the hyperparameters such as learning rates and observables following the general practice in Jerbi et al. [11], Ding and Spector [15], which are also summarized in Table 1. All the agents are trained using REINFORCE [43], which is a basic Monte Carlo policy gradient algorithm. We apply the value-function baseline [44] in MountainCar to stabilize the Monte Carlo process, which has been commonly used in recent RL methods [39].

*Time Complexity Analysis.* Since the search runs for 50 generations with a population size of 30, and we reduced the number of episodes to a factor of 0.5, the theoretical time complexity upper bound of the search process is 750× the runtime of training one single PQC model. In practice, we find our method takes only around 200× the cost of training an alternating-layer PQC, due to the following factors: (1) a reduced number of episodes does not affect the evaluation for selection in GAs but can significantly reduce the training time, because later episodes are usually close to the optimal solution, meaning the RL agents will likely keep running for a longer period; (2) some of the architectures in the search space are either too small or not effective, taking much less time to train and evaluate.

We also use parallelization in the search process, which further reduces the clock time to around 40× training a single alternating-layer PQC, i.e., around 2 days on a 64-core machine. More specifically, for each generation, all the candidate architectures are trained and evaluated simultaneously. All the experiments were run on a computing cluster with up to 64 cores per job. The total estimated compute time of all the experiments in this work is 30,000 CPU hours.

### 4.3. Single-Objective Results

First, we evaluate the learning performance of the proposed MEAS-PQC by training with a single objective of average rewards. The purpose of this experiment is to show the effectiveness of variable-length genome representation in MEAS-PQC by using UMAD. We compute and visualize the average learning performance over 20 trials of the best-performing architectures searched by MEAS-PQC and compare to recent work (EQAS-PQC [15] and Softmax-PQC [11]). To ensure a fair comparison, for Softmax-PQC, we use the depth of 5, resulting in an architecture with length 16, which is greater than the resulting architectures searched by MEAS-PQC for all the environments. The learning performance is represented as the area under the learning curve, which is identical to the average collected reward. As shown in Figure 2, we can see that our method is able to find PQC architectures that outperform both EQAS-PQC and the standard alternating-layer PQC by a significant margin. While the final rewards may converge to optimum for all the methods (especially for easy RL environments like CartPole), the PQC architectures searched by our method is able to learn much quicker in early episodes.

### 4.4. Multi-Objective Results

In Section 3.4, we propose three objectives: learning performance, quantum noise, and model size. Given the objective functions, we test the proposed method under different multi-objective settings. More specifically, 1-obj is optimizing using only the learning performance (average reward); 2-obj is optimizing both learning performance and quantum noise; 3-obj is optimizing all three objectives. The results are shown in Table 2.

We can observe that first, the proposed MEAS-PQC (1-obj) outperforms other methods by a significant margin with architectures that have less or comparable quantum noise and model size. For different multi-objective settings, we can observe that both MEAS-PQC (2-obj) and MEAS-PQC (3-obj) have reduced quantum noise and model size comparing to MEAS-PQC (1-obj), while maintaining similar performance. By using multi-objective optimization, the proposed method is able to search for architectures not only have good learning performance, but also optimized for real-world computation needs such as quantum-noise tolerance and model size.

### 4.5. Pattern Analysis in Quantum Architectures

To further interpret the results and discover the performance-critical design choices of PQCs as opposed to the commonly-used alternating-layer architecture, we perform qualitative analysis on the top-performing architectures searched by MEAS-PQC. We perform pattern mining on the architectures and calculate the frequency of patterns with different lengths of 2, 3, 4, and 5. The results are shown in Table 3, and we can observe that consecutive $x_{3}$ and alternating $x_{1}$, $x_{2}$ are the most common patterns, which indicates that there may exist essential substructures of quantum circuits that lead to good learning performance.

In addition, we also calculate and visualize the probability distribution of each quantum operation over the positions in the architecture, as shown in Figure 3. We can observe that the V-PQC has a similar frequency as entanglement, which aligns with the alternating-layer design. However, the frequency of D-PQC has an obvious decreasing trend, indicating that it is better to have more Data-encoding PQCs at the beginning of the architecture. This finding further indicates that more classical computation is likely to be needed in the shallow part of the hybrid quantum learning models.

## 5. Discussion of Limitations and Broader Impacts

There are several limitations to our work. First, due to framework constraints, all the experiments are conducted using a simulation backend, and the quantum noise and computation cost are evaluated based on the proposed theoretical metrics. Second, for generality, this work considers genome abstractions for more structural quantum operations, e.g., qubit rotation in 3-dimensional space and circular entanglement. It is easy to extend our work to have a much larger search space by using single-dimensional rotations and partial entanglement, and we look forward to exploring these extensions in future work. The primary goal of this work is to show that the proposed method can automate and improve the macro architecture design of PQC-based hybrid quantum-classical learning systems, with a fair comparison to prior work.

In general, we expect our work to have positive societal impacts. While prototyped in a simulated environment, the proposed method could be beneficial to near-term quantum computing applications, especially quantum RL. Since modern RL tasks usually require huge amounts of computation, our method may provide significant computational advantages regarding both time and energy consumption. Our work further considers constraints such as quantum noise and model size, which could make the proposed method more pragmatic for real-world applications.

On the other hand, like other recent work in RL and machine learning research in general, our work could have negative societal impacts. Concerns such as whether RL applications will have positive impact on the society are naturally inherited by our work. More specifically for quantum RL, while we made an effort to explain the results, the overall interpretability of quantum systems is still not as good as that of many classical systems, which may lead to negative consequences. Moreover, the requirement for quantum computing hardware to obtain computational benefits may increase inequality because of unequal access to quantum computing resources.

## 6. Conclusions

In this work, we propose MEAS-PQC, a multi-objective evolutionary quantum architecture search framework for hybrid quantum learning systems. MEAS-PQC uses a population-based genetic algorithm with UMAD and quantum-specific configurations to evolve PQC architectures by exploring the search space of quantum operations. Experimental results show that our method can significantly improve the learning performance of PQC-based systems in solving benchmark reinforcement learning problems. The results on multi-objective optimization further show that our method can search for architectures that have good learning performance and are also optimized for reduction in quantum noise and model size. We also perform analysis to extract and interpret the performance-critical architecture design choices. In future work, we expect to apply our method on real quantum hardware and use hardware-specific metrics for objectives to show the performance in a more pragmatic setting.

## Figures and Tables

**Figure 1 entropy-25-00093-f001:**
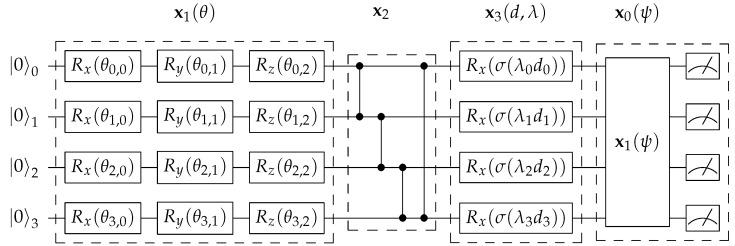
Illustration of a simple 4-qubit architecture in the MEAS-PQC search space. The architecture (with genome encoding 1−2−3−0) is composed of 4 operations: (1) Variational PQC ($x_{1}$) performs rotations on each qubits according to parameters θ; (2) Entanglement ($x_{2}$) performs circular entanglement to all the qubits; (3) Data-encoding PQC ($x_{3}$) performs rotations on each qubit according to the input data *d*, scaling parameter λ, and activation function σ; (4) Measurement ($x_{0}$) adds another Variational PQC ($x_{1}$) and performs measurement to obtain the observable values.

**Figure 2 entropy-25-00093-f002:**
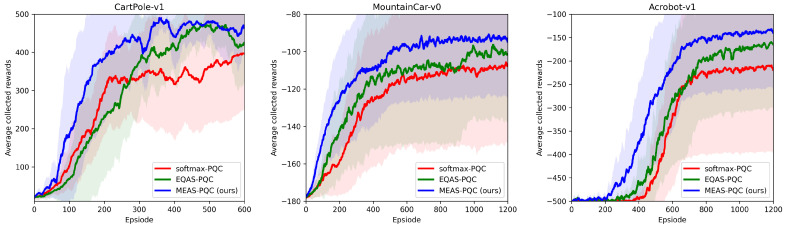
Learning performance on benchmark RL environments. We plot the learning curves (smoothed by a temporal window of 10 episodes) averaged over 20 trials of the resulting MEAS-PQC architecture compared against EQAS-PQC [15] and Softmax-PQC [11] on three benchmark RL environments. The shaded areas represent the standard deviation of the average collected reward.

**Figure 3 entropy-25-00093-f003:**
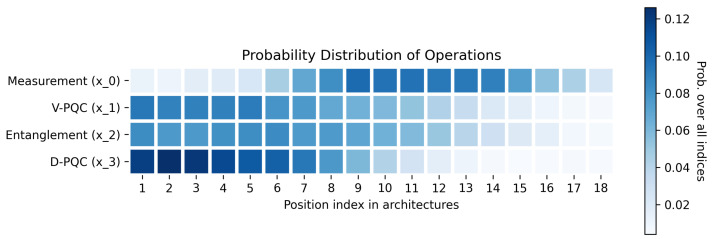
Probability distribution of quantum operations over positions in top-performing PQC architectures. We select 30 top-performing architectures searched by MEAS-PQC (10 for each RL environment), and visualize the probability distributions of each operation over its positions.

**Table 1 entropy-25-00093-t001:** RL environment specifications and hyperpameters. (*: The reward function of MountainCar-v0 has been modified from the standard version in OpenAI Gym, following the practices in Jerbi et al. [11], Duan et al. [39].)

Environment	# States/Qubits	# Actions	Reward	γ	Horizon	Episodes
CartPole-v1	4	2	+1	1.0	500	600
MountainCar-v0	2	3	$-1+height{\;}^{*}$	1.0	200	1200
Acrobot-v1	6	3	−1	1.0	500	1200
**Environment**	**Learning Rates ($\alpha_{\theta }$, $\alpha_{w}$, $\alpha_{\lambda }$)**		**Observables**	**Value-Function Baselines**		
CartPole-v1	0.01,0.1,0.1		[$Z_{0}Z_{1}Z_{2}Z_{3}$]	None		
MountainCar-v0	0.01,0.1,0.01		[$Z_{0},Z_{0}Z_{1},Z_{1}$]	Linear baseline [39]		
Acrobot-v1	0.01,0.1,0.01		[$Z_{0},\cdots ,Z_{5}$]	Linear baseline [39]		

**Table 2 entropy-25-00093-t002:** Multi-objective Evaluation. For each method, we evaluate the performance based on the three objectives: learning performance, quantum noise, and computation cost, as described in Section 3.4. The results show that, with multi-objective optimization, MEAS can produce architectures with less quantum noise and computation cost, while still maintaining superior learning performance.

Environment	Objectives	Softmax-PQC [11]	EQAS-PQC [15]	MEAS-PQC (Ours)		
				1-obj	2-obj	3-obj
CartPole	Avg. Reward	271.0 ± 78.8	317.6 ± 67.7	349.8 ± 71.3	**351.6** ± **64.8**	342.3 ± 76.2
	Quantum noise	30	20	25	17	**15**
	Model size	20	**12**	19	13	13
MountainCar	Avg. Reward	−126.6 ± 33.4	−119.5 ± 32.8	**−108.2** ± **25.8**	−110.7 ± 21.4	−111.3 ± 25.2
	Quantum noise	30	21	20	**14**	17
	Model size	20	13	14	**8**	11
Acrobot	Avg. Reward	−353.0 ± 93.7	−328.5 ± 74.6	**−280.7** ± **79.3**	−283.4 ± 78.6	−289.3 ± 75.5
	Quantum noise	30	17	22	19	**15**
	Model size	20	13	16	**11**	**11**

**Table 3 entropy-25-00093-t003:** Pattern frequency in top-performing PQC architectures. We select 30 top-performing architectures searched by MEAS-PQC (10 for each RL environment), and calculate the frequency of patterns with different lengths. We can observe that consecutive $x_{3}$ and alternating $x_{1}$, $x_{2}$ are the most common patterns.

len-2		len-3		len-4		len-5	
Pattern	Freq.	Pattern	Freq.	Pattern	Freq.	Pattern	Freq.
(3, 3)	0.282	(3, 3, 3)	0.213	(3, 3, 3, 3)	0.122	(1, 2, 1, 2, 1)	0.087
(2, 1)	0.245	(1, 2, 1)	0.184	(2, 1, 2, 1)	0.122	(1, 3, 3, 3, 3)	0.075
(1, 2)	0.205	(2, 1, 2)	0.142	(1, 2, 1, 2)	0.102	(2, 1, 2, 1, 2)	0.069
(1, 3)	0.099	(2, 1, 3)	0.067	(1, 3, 3, 3)	0.068	(3, 3, 3, 3, 2)	0.064
(3, 2)	0.066	(3, 2, 1)	0.059	(3, 3, 3, 1)	0.063	(3, 3, 3, 3, 1)	0.064

## Data Availability

The data presented in this study are openly available via OpenAI Gym [19]: https://www.gymlibrary.dev/.

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
