# Peer review of "Multi-Objective Evolutionary Architecture Search for Parameterized Quantum Circuits"

_entropy, 2023, doi:10.3390/e25010093_

Round 1
Author Response
Dear Reviewer,
Thanks for the comprehensive review. Please find below our response to address your questions:
1) "The introduction given is too long for the total length of the paper"
The introduction part is around 1 page out of 11 pages of the paper, and we thought that was a proper length. But anyway, we make it briefer in the revision and hope that will help with readability.
2) "The introduction given covers only the very basics of quantum computing, not even more than one example of entangling gates. In terms of clarity, it is quite difficult for me to follow what is the meaning of many of the concepts appearing in the paper."
We covered many related terms and basics in Sec. 2.1 and 2.2. In the revision, we added a few brief descriptions of terms in the introduction and refer the readers to the following sections for more detailed descriptions. We also added a few more basics in Sec. 2.1 and 2.2.
3) "For example, the term PQC is something very general that can be defined as "all possible combinations of trainable and non-trainable gates forming a common operation". However, in this case, PQC’s are just sets of single-qubit operations (apparently), which contradicts the intuition of Figure 1 - ?0, for example."
At the beginning of the paper, we stated that our scope is using PQC as machine learning models [3], where PQCs are typically composed of fixed gates, e.g. controlled NOTs, and adjustable gates, e.g. qubit rotations. In our search space design, we try to follow some successful practices from prior work such as using single-qubit gates (rotations) and multi-qubit gates (controlled z for entanglement). In the revision, we make this point clear by adding a few descriptions in Intro and Method.
4) "In particular, having only one entangling layer in the pool limits dramatically the performance of this algorithm. Without that, the work essentially falls back to be practically equal to the one in [1] "
No, while the pool only has one entangling layer, it can appear anywhere in the architecture multiple times. We describe the search space of our method in Sec. 3.1, especially Eq. 7.
5) "what is exactly the difference between the present paper and [1]. As I understand, the contribution is the use of new optimizers, in particular the multi-objective one, to solve the problems."
In this work, we propose an evolutionary method to search for good PQC architectures. [1] only uses a fixed architecture, which is similar to the one we show in Fig. 1 as an example. Our method searches for optimal architecture based on performance, and we use multi-objective optimization to solve for both performance and the cost of PQC. In the revision, we added more comparisons between our method and others in related work.
6) "The objective functions to minimize seem arbitrary to me, since in fact noise and model size are just ways of counting the number of gates coming into the circuit. This can in principle be optimized by adding this quantity into the first reward."
No, while you may theoretically add the objectives together, you need to carefully weigh each objective for each RL problem, because the scales of RL rewards are different. Our method handles this situation. We added the description to this point in the revision.
7) "A look into table 2 reveals that the claim of superior learning performance holds for MEAS-PQC with respect to the other options, but the multi-objective optimization is not necessarily a requirement. In fact, I do not understand the meaning of MEAS-PQC for the case of one objective, if by definition the model is designed for multi-objective optimization. "
So without multi-objective optimization, our method is only searching for PQC architectures that have good performance, which may be costly computation-wise. As shown in Table 2, the resulting PQC architectures for 1-obj always have larger model sizes and quantum noise.
8) "Also, the uncertainties are so large that it is almost impossible to conclude any advantage. "
This is due to the nature of RL problems, since good and bad solutions may have very different rewards. We follow the common practice to report the average performance of 20 trials, which is usually sufficient to conclude an advantage in the RL literature [4].
9) "First, one could program the agent to avoid repetition, so that this fusing is not required, and second, the problem of compiling a circuit into shallower pieces is very hard."
For the first, this is equivalent to what we are currently doing, since we perform duplicate elimination at each generation before evaluation. For the second, since the compiled circuit is still in the search space of our method, it is equivalent to just using that shallow circuit, which adds no extra difficulty to generating that circuit.
10) "Finally, I found the paper in [2] to be extremely similar to this one. "
[2] is a pilot study of this work, which was submitted to a workshop for discussion and improvement (i.e., not counted as a complete publication). This paper makes a significant improvement on [2] by including 1) an enhanced evolutionary method to navigate the search space with a better mutation method and duplicate elimination; 2) new objectives and experiments on multi-objective optimization. Both Table 2 and Fig. 2 show our improvement over [2]. In the revision, we added a paragraph in related work to highlight the improvement of this work.
11) "As a summary, there are two main questions to answer and solve in order to consider this paper for publication, which are a) why is this work consistently different from [1] or [2], and b) why is the multi-objective optimization required. "
For a), please see 4), 5), and 10). For b), please see 6) and 7).
References
[1] Andrea Skolik, Sofiene Jerbi, and Vedran Dunjko. Quantum agents in the gym: a variational quantum algorithm for deep q-learning. Quantum, 6:720, may 2022.
[2] Li Ding and Lee Spector. Evolutionary quantum architecture search for parametrized quantum circuits. In Proceedings of the Genetic and Evolutionary Computation Conference Companion. ACM, jul 2022.
[3] Benedetti, M.; Lloyd, E.; Sack, S.; Fiorentini, M. Parameterized quantum circuits as machine learning models. Quantum Science 358 and Technology 2019, 4, 043001.
[4] Brockman, G.; Cheung, V.; Pettersson, L.; Schneider, J.; Schulman, J.; Tang, J.; Zaremba, W. Openai gym. arXiv preprint 402 arXiv:1606.01540 2016.
Reviewer 2 Report
The authors propose a new approach (MEAS-PQC) to search for optimal parametrized quantum circuits (PQC). They test the performance of their proposed method in three scenarios. This work is important as plenty of applications (for example, VQE applied to finding ground state energies of molecules) rely on finding shallow yet ‘good’ PQCs, but little is known or understood about PQCs.
The authors motivate the work well in the introduction, and the background theory is explained in good detail. Their methods and results sections are reasonably clear too. In summary, the manuscript is well written.
I recommend the manuscript for publication with some changes.
Some quick questions/comments:
-‘Near-term quantum computing technologies empower quantum computing systems to reliably solve tasks that are beyond the capabilities of classical systems’:
I’m not sure if this sentence is required. Quantum advantage has only been shown for very few problems like pseudo-random number generation and boson sampling. I would expect that for most useful problems, devices from NISQ era may not show an advantage.
-‘If a quantum state of a composite system can not be written as a product of the states of its components, ‘:
‘Tensor product’ instead of ‘product’ may be better.
-‘An entanglement can be created by applying controlled-Pauli-Z gates to the input qubits. ‘:
Maybe it is better to tell ‘by applying appropriate 2-qubit gates’ instead of ‘controlled-Pauli-Z gates’. This is because CX can generate entanglement too, for example. In fact, it is far easier to generate entanglement using CX since it can flip 1 to 0 and 0 to 1 on the target qubit when control is 1, whereas CZ can only introduce a phase.
-Fig 1: Just out of curiosity, is there any specific reason to choose 3 rotations (except generality, as mentioned in the manuscript)? Would not just RyRz be sufficient for each qubit, in that for an arbitrary combination of the two angles, one covers the entire Bloch sphere?
-Section 3.1 on encoding scheme and search space: I personally felt that this could have been discussed in a bit more detail. Especially the bullet point on measurement was a bit too short, I felt.
-I also felt that the authors could add one short para on the connection between genetics and the simple PQC in Fig 1, for clarity for the readers.
-Equation (11): any qualitative justification for this ‘ansatz’?
-Fig 2: I would suggest that the authors use red, green, and blue, for example, for the three curves, just to make sure that readers with color perception issues have no problem appreciating the results.
-Fig 3: please represent x_0, for example, with 0 as a subscript, etc.
-Table 1: I think some more explanation needs to be provided, by perhaps taking one environment as a representative example (say, CartPole-v1), and then:
a. comment on the choice of qubit (what is chosen as qubit in the example? Why is it a qubit, in the sense that what then plays the role of |0> and what is |1>?,
b. details such as for what example case and why the authors chose 4 qubits,
c. when the authors choose 4 qubits for, for example, the CartPole-v1 problem, and begin with a population size of 30, did they choose circuits like the one in Fig 1? What are the update rules from one generation to another? What would the length of the architecture = 20 mean here? , etc.
This will help the readers connect the quantum part and the CartPole part (which is a classical problem), and appreciate the importance of the results shown by the authors.
-In ‘Time complexity analysis’, the authors can maybe give other details (besides the number of cores) such as CPU RAM, GPU (if it has been used), etc.
Author Response
Dear Reviewer,
Thanks for the comprehensive and detailed review. Please find below our response to address your questions:
1) "Quantum advantage has only been shown for very few problems like pseudo-random number generation and boson sampling. I would expect that for most useful problems, devices from NISQ era may not show an advantage. "
Thanks for your suggestion. We agree with the point that currently there has not been a clear example where NISQ devices will show quantum advantage in general quantum machine learning. However, recent work [1] constructed RL environments with a provable separation in learning performance between quantum and classical methods. Considering the possibility that some existing problems may be formed in such ways, we positively think that NISQ devices will have a big impact on Quantum ML. In the revision, we rephrase the paragraph to make it more concise.
2) "‘Tensor product’ instead of ‘product’ may be better. "
Thanks, and we make the changes in the revision.
3) Maybe it is better to tell ‘by applying appropriate 2-qubit gates’ instead of ‘controlled-Pauli-Z gates’. This is because CX can generate entanglement too, for example. In fact, it is far easier to generate entanglement using CX since it can flip 1 to 0 and 0 to 1 on the target qubit when control is 1, whereas CZ can only introduce a phase.
Right, we specifically stated CZ because, in this work, we did not consider other ways to generate entanglement. We modified the paragraph according to your suggestion in the revision.
4) Fig 1: Just out of curiosity, is there any specific reason to choose 3 rotations (except generality, as mentioned in the manuscript)?
There has been some prior work using 3 rotations (also a few using 2 rotations). So we are just going with 3 for generality, but it would be interesting to see which one works better in practice for QML, and we look forward to exploring that in future work.
5) "-Section 3.1 on encoding scheme and search space: I personally felt that this could have been discussed in a bit more detail. Especially the bullet point on measurement was a bit too short, I felt. "
We expanded on this section to include more descriptions.
6) "-I also felt that the authors could add one short para on the connection between genetics and the simple PQC in Fig 1, for clarity for the readers. "
We added a short paragraph in the Sec. 3.1 to include more basics for genetic algorithms.
7) -Equation (11): any qualitative justification for this ‘ansatz’?
Our idea is to assume that the total quantum noise is proportional to the total number of quantum gates operating on any qubit, as we described in Sec. 3.3. However, since the errors of different quantum gates and operations have not been found in past empirical studies, we propose this artificial measure as a proof-of-concept for our multi-objective optimization. We make this point clear in the revision.
8) -Fig 2: I would suggest that the authors use red, green, and blue, for example, for the three curves, just to make sure that readers with color perception issues have no problem appreciating the results.
Thanks for your suggestion, and we will make changes accordingly.
9) -Table 1: I think some more explanation needs to be provided, by perhaps taking one environment as a representative example (say, CartPole-v1), and then:
a. comment on the choice of qubit (what is chosen as qubit in the example? Why is it a qubit, in the sense that what then plays the role of |0> and what is |1>?,
We follow prior work to use each qubit to represent a state variable. There is no actual meaning of |0> or |1>, but the data-encoding PQC will rotate the qubit based on the value of the state variable. We will add some explanations in Sec. 4.1.
b. details such as for what example case and why the authors chose 4 qubits,
As explained above, the number of qubits is just the number of state variables. We did not further optimize the number of qubits in the scope of this work, but that can certainly be done with the proposed method, by including the size of qubits in the search space.
c. when the authors choose 4 qubits for, for example, the CartPole-v1 problem, and begin with a population size of 30, did they choose circuits like the one in Fig 1? What are the update rules from one generation to another? What would the length of the architecture = 20 mean here? , etc.
This will help the readers connect the quantum part and the CartPole part (which is a classical problem), and appreciate the importance of the results shown by the authors.
We add a few more details in the experiment results part, but most of these questions have been described in the paper in different places. More specifically, the initial architectures are sampled randomly; the update rules using the genetic algorithm, NSGA-II (and we added more descriptions); architecture=20 means there are 20 genomes (x0-x3) in the architecture.
10) -In ‘Time complexity analysis’, the authors can maybe give other details (besides the number of cores) such as CPU RAM, GPU (if it has been used), etc.
Only the CPU is used, and the experiments just need normal RAM.
[1] Jerbi, S.; Gyurik, C.; Marshall, S.; Briegel, H.; Dunjko, V. Parametrized Quantum Policies for Reinforcement Learning. Advances in 373 Neural Information Processing Systems 2021, 34.
Round 2
Reviewer 1 Report
Dear authors,
Thanks for addressing the issues I raised with respect to this paper. Let me go for the replies you gave me.
1) Maybe I was not clear enough, but one of my major issues was related to the amount of detail given in the introduction with respect to the rest of the paper. Such introduction left me expecting something in more detail. In any case, I acknowledge the consideration on brevity.
2) For the basic concepts in QC, I still think that there is a gap between the mentioned concepts and the overall goal of the problem, and the content did not change considerably. However, this is still no founded reason to reject the work.
3) I do not find the change substantial as to change my personal opinion on this statement. There are still many details that could be discussed, for example that encoding gates come with some tunable weight. This concept arises from the paper in re-uploading schemes, and also in Schuld's paper, but has been later overcome.
4) I am still convinced that the pool is limited. In fact, any repetition of such layer is the same as a reordering of qubits, probably with less power. One could choose to use for example pairing entangling gates, or sparse, but in this case the choice is forced from the method and is just having the whole chain of entangling layers. Even though this is a valid choice, it would be nice to mention it. The freedom in where it appears is not enough to justify this.
5) Incremental improvement then
6) Agree with this point, but is there any reason why the naive addition would fail?
7) Understood. However I want to add that my understanding of it (in a quick reading) was that you were using a single function (as an addition for instance) of the different values, not that only the Avg. reward was considered. I suggest adding this info in the caption for clarity.
8) I am no expert in RL literature, but these uncertainties still sound very large to me. There is no clear separation.
9) First: it is equivalent from a theoretical point of view, but not practical. Adding this extra constraint from the beginning could avoid a complicated optimization. Second, agreed, but then the problem you face is how to create a shallow and still competitive circuit.
10) Even though I understand the relationship, I still think that the two papers are too similar. A plagiarism detector would set an alarm to it. I do not have any problem with it, but the editor may raise it as a significant issue depending on the policies. I would like to deflect this question to the journal.
Hope this second round is also useful for improving the paper.
Author Response
Dear Reviewer:
Thanks for the follow-up. Please see below our responses.
3) There are still many details that could be discussed ... (regarding PQCs).
We added more details in Sec. 2.2 to discuss different PQCs and how they are optimized in the scope of quantum learning. Since they are basically just single-qubit rotations, we think the current descriptions (in both Sec. 2.2 and Sec. 3.2) are quite sufficient.
4) I am still convinced that the pool is limited.
We state the limitation of search space in Sec. 5. While we did not explore more interesting combinations of quantum operations in this work as mentioned by the reviewer, current results already show significant benefits we can obtain from using the proposed framework, and reveal the potential of our method if extended to larger search space.
5) Agree with this point, but is there any reason why the naive addition would fail?
The main disadvantages of naive addition (i.e., Weighted Sum Method) are: 1) It is difficult to manually set the weight vectors to obtain a Pareto-optimal solution in the desired region in the objective space (as I mentioned in the previous reply); 2) It cannot find certain Pareto-optimal solutions in the case of a nonconvex objective space. (And there is no guarantee that our QAS search space is convex.) Multi-objective optimization (MOO) has been an active field of research in the past two decades and we refer the reviewer to some readings[1, 2] for the fundamentals of MOO. We also added these references to the paper in the revision.
[1] Gunantara, N., A review of multi-objective optimization: Methods and its applications. Cogent Engineering, 5(1), 1502242, 2018
[2] K. Deb, Multi-Objective Optimization using Evolutionary Algorithms, John Wiley & Sons, Inc., 2001
9) First: it is equivalent from a theoretical point of view, but not practical. Adding this extra constraint from the beginning could avoid a complicated optimization. Second, agreed, but then the problem you face is how to create a shallow and still competitive circuit.
First: our duplication elimination does serve as a constraint before training the architecture. Second: Eq. 11 and 12 show how we create a shallow equivalent of complex circuits.
10) Even though I understand the relationship, I still think that the two papers are too similar.
We made more changes in phrasing in the revision, and we would like the editor to comment on this if there is a problem.
Thanks for your time and please kindly let us know if you have further comments and suggestions.
Reviewer 2 Report
I recommend the manuscript in its current form for publication.
Author Response
We would like to thank the review for their recommendation.